# REGULARIZED-OFU: AN EFFICIENT ALGORITHM FOR GENERAL CONTEXTUAL BANDIT WITH OPTIMIZATION ORACLES

## ABSTRACT

In contextual bandit, one major challenge is to develop theoretically solid and empirically efficient algorithms for general function classes. We present a novel algorithm called *regularized optimism in face of uncertainty (ROFU)* for general contextual bandit problems. It exploits an optimization oracle to calculate the well-founded upper confidence bound (UCB). Theoretically, for general function classes under very mild assumptions, it achieves a near-optimal regret bound $\tilde{O}(\sqrt{T})$. Practically, one great advantage of ROFU is that the optimization oracle can be efficiently implemented with low computational cost. Thus, we can easily extend ROFU for contextual bandits with deep neural networks as the function class, which outperforms strong baselines including the UCB and Thompson sampling variants.

## 1 INTRODUCTION

Contextual bandit is a basic sequential decision-making problem which is extensively studied and widely applied in machine learning. At each time step in contextual bandit, agent should choose an action according to a presented context, and will receive a reward conditioned on the context and the selected action. The goal of the agent is to maximize its cumulative reward, which is equivalent to minimizing regret.

Algorithms for contextual bandit can be divided into two categories: *agnostic algorithms* and *realizability-based algorithms*. The agnostic algorithms, e.g., EXP4 (Auer et al., 2002b; McMahan & Streeter, 2009; Beygelzimer et al., 2011), provide worst-case optimal regret bounds for any function class and data. However, the time complexity of EXP4 is linear to the cardinality of the function class which is intractable for large function classes.

The *realizability* assumes that the reward is generated from an underlying model, whose form is known but with some parameters to be determined. When the realizability is satisfied in real-world problem, the realizability-based algorithms usually perform much better than the agnostic algorithms. The most popular realizability-based algorithms are UCB (Auer et al., 2002a) which selects action according to an upper confidence bound, and Thompson sampling (Thompson, 1933) which makes decisions according to samples from the posterior distribution. Both UCB and Thompson sampling achieve near-optimal regret bound for many function classes. However, the construction of upper confidence bound and sampling from the posterior distribution are extremely computationally expensive for general function classes.

To overcome the computational barrier, there are a line of works (Agarwal et al., 2014; Dudik et al., 2011; Foster & Rakhlin, 2020; Foster et al., 2018) that reduce the decision-making problem to an optimization problem, and then exploit optimization oracles to accelerate computation. Nonetheless, the optimization oracle may not be feasible or efficient for general function classes. In face of the challenges above, when dealing with modern function classes, such as deep neural networks, these theoretically solid algorithms either become computationally intractable or do not achieve low regret in practice.

Our paper also falls into this line of research. We propose a novel algorithm, called Regularized OFU (ROFU) which is developed upon the realizability assumption and an optimization oracle. In ROFU,

we measure the uncertainty of the reward function by a regularizer, and then calculate the optimistic estimation by maximizing the reward function with the regularizer penalizing its uncertainty. To the end, we give a novel formulation of upper confidence bound for general function classes. Our algorithm achieves near-optimal regret bound under very mild assumptions, and can be efficiently solved by standard optimization oracles, such as gradient descent. Thus, our algorithm can be easily extended to deep neural networks in a computationally efficient manner.

We summarize our contributions as follows:

- We propose a new UCB variant ROFU, which is designed for general function classes with provably near-optimal regret. ROFU computes UCB with an optimization oracle which can be efficiently implemented for complex function classes including deep neural networks.
- Theoretically, for general function classes under very mild assumptions, we prove that ROFU achieves a regret of $O(\sqrt{T \log \frac{|\Theta|T}{\delta}})$ which matches the lower bound up to a logarithm factor, where $\Theta$ is the parameter space. As a special case, we present a regret bound for linear function class which is the same to that in (Abbasi-Yadkori et al., 2011).
- Empirically, we evaluate ROFU on complex contextual bandits with deep neural network as the function class. We show that ROFU also provides efficient UCB estimation for popular DNN architectures including MLP and ResNet. Moreover, our algorithm enjoys a smaller regret than strong baselines on real-world non-linear contextual bandit problems introduced by Riquelme et al. (2018).

## 2 PRELIMINARY

We consider the contextual bandit problem (Bubeck & Cesa-Bianchi, 2012) with $K$ actions.

**Definition 1** (Contextual bandit). *Contextual bandit is a sequential decision-making problem where the agent has a set of actions $A$. At each time step $t$, the agent first observes a context $x_t$, then selects an action $a_t \in A$ based on the context. After taking the action, the agent receives a reward $r_t$.*

Realizability-based algorithms are developed under the following assumption.

**Assumption 1** (Realizability assumption). $\mathbb{E}r_t := f_{\theta^*}(x_t, a_t)$ *where* $f_{\theta^*}(x_t, a_t)$ *is a function with unknown groundtruth parameters* $\theta^* \in \Theta$.

The agent aims to maximize its expected cumulative reward $\sum_{t \leq T} f_{\theta^*}(x_t, a_t)$ which is equivalent to minimizing the regret $R_T = \sum_{t \leq T} \max_a (f_{\theta^*}(x_t, a) - f_{\theta^*}(x_t, a_t))$ under the realizability assumption. For convenience, let $a_t^* = \arg\max_a f_{\theta^*}(x_t, a)$.

## 3 METHOD

---
**Algorithm 1** Regularized Optimism in Face of Uncertainty
---
1: **Input**: A reward function $f_{\theta^*}$ with unknown $\theta^*$, number of rounds $T$.
2: $D_0 := \emptyset$.
3: **for** t = 1, ..., T **do**
4:     Observe $x_t$.
5:     $\forall a \in A$, compute

$$OFU^R(x_t, a) = \begin{cases} U(x_t, a) & \text{Option I for general function classes} \\ \text{Lin-U}(x_t, a) & \text{Option II, an improved version for linear functions} \end{cases} \quad (1)$$

6:     Take $a_t = \arg\max_{a \in A} OFU^R(x_t, a)$ and receive reward $r_t$ with $\mathbb{E}r_t = f_{\theta^*}(x_t, a_t)$.
7:     Let $D_t := D_{t-1} \cup \{(x_t, a_t, r_t)\}$.
8: **end for**
---

In this section, we first formally present the optimization oracle and the algorithm. Then we discuss the intuition behind our method. After that we provide the theoretical justification of the algorithm,

showing it's near-optimal in terms of regret under very mild assumptions. Finally, we give an empirically efficient implementation of the algorithm relying on gradient descent. Proofs of our theoretical results can be found in Section 3.3 and Appendix.

Our algorithm is developed upon the following oracle.

**Definition 2** (Optimization oracle). *Given dataset $D := \{(x_{t'}, a_{t'}, r_{t'})\}_{t' < t}$ before round t, for $x_t, a$, we assume there is an optimization oracle to compute*

$$U(x_t, a) = \max_\theta f_\theta(x_t, a) - \eta_{t,a}(MSE(\theta; D) + \alpha \|\theta\|^2), \tag{2}$$

*where $MSE(\theta; D) = \frac{1}{|D|} \sum_{(x,a,r) \in D} (f_\theta(x, a) - r)^2$ and $\eta_{t,a}, \alpha > 0$ are constants to be specified later. For convenience, let $\hat{\theta}_{t,a} = \arg\max_\theta f_\theta(x_t, a) - \eta_{t,a}(MSE(\theta; D) + \alpha \|\theta\|^2)$.*

The availability of such optimization oracle is a very mild assumption in practice. For example, we can exploit gradient-based algorithm to approximately solve Eq. (2) for differentiable functions.

As summarized in Alg. 1, our algorithm is as follows: in round $t$, the agent invokes the optimization oracle to compute $\text{OFU}^R(x_t, a)$ for each action. Then, the agent selects $a_t = \arg\max_a \text{OFU}^R(x_t, a)$.

We now provide more insights into Alg. 1 and the optimization oracle in Eq. (2). The key to minimize regret is to trade-off exploration and exploitation (EE). In order to maximize cumulative reward, the agent exploits collected data to take the action with high estimated reward while it also explores the undiscovered areas to learn knowledge. Our method follows the *Optimism in Face of Uncertainty (OFU)* principle, which is widely verified to be effective in EE trade-off. When facing uncertainty, OFU first optimistically guesses how good each action could be and then takes the action with highest guess.

Eq. (2) gives such an optimistic guess by maximizing $f_\theta(x_t, a)$ under the regularization of mean squared error. The intuition behind Eq. (2) is that: from the view of exploitation, if $\hat{\theta}_{t,a}$ is a parameter with a small mean squared error, and $f_{\hat{\theta}_{t,a}}(x_t, a)$ is large. Then we can expect $f_{\theta^*}(x_t, a)$ is also large as $\hat{\theta}_{t,a}$ is close to $\theta^*$ in general; from the view of exploration, if $\hat{\theta}_{t,a}$ increases the value of $f_{\hat{\theta}_{t,a}}(x_t, a)$ without significantly increasing $\text{MSE}(\hat{\theta}_{t,a}; D)$, then the uncertainty on the reward of $(x_t, a)$ would be large.

## 3.1 Regret analysis

Besides the conciseness and clear intuitions, the algorithm also enjoys several theoretical advantages: Theorem 1 develops a $O(\sqrt{T \log \frac{|\Theta|T}{\delta}})$ regret under Assumption 2. The regret bound in Theorem 1 matches the lower bound as presented in Theorem 2 up to a logarithm factor, showing the algorithm is near-optimal.

**Assumption 2.** *There exists constants $c_1, c_2 > 0$ and a function $g : \Theta \to \mathbb{R}^+ \cup \{0\}$, such that*

$$\forall x, a, \theta, |f_\theta(x, a) - f_{\theta^*}(x, a)| \in [c_1 g(\theta), c_2 g(\theta)].$$

Assumption 2 is very mild that includes many function classes. For example, if $\forall x, a$, function $h_{x,a}(\theta) = |f_\theta(x, a) - f_{\theta^*}(x, a)|$ is $c_1$-strongly convex and $c_2$-smooth, then Assumption 2 is true.

**Theorem 1** (Regret for general functions with Option I). *Under Assumption 2, if $\forall \theta, x, a, f_\theta(x, a) \in [-1, 1]$, let $\eta_{t,a} = \frac{c_2}{8c_1} \sqrt{\frac{t-1}{\log \frac{|\Theta|T}{\delta}}}$ and $\alpha = 0$, then with probability at least $1 - \delta$, the regret of Alg. 1 with Option I satisfies*

$$R_T \leq 8 \frac{c_2}{c_1} \sqrt{T \log \frac{|\Theta|T}{\delta}}. \tag{3}$$

**Theorem 2** (Lower bound). *For any $c_2 \geq c_1 > 0$, there is a function class $\{f_\theta\}_{\theta \in \Theta}$ that satisfies Assumption 1 and 2 with some $g(\theta)$. One can construct a context sequence $\{x_t\}_{t \leq T}$ such that the expected regret of all bandit algorithms is lower bounded by $\Omega(\sqrt{T \log |\Theta|})$.*

It is obvious that when $c_2/c_1$ is large, the regret bound in Theorem 1 is meaningless. However, fortunately, for linear functions, which is the most interesting function class with $c_2/c_1 = \infty$, Alg. 1 achieves a near optimal regret bound as in Theorem 3 and 4, indicating boarder theoretical potentials to develop regret bound of Alg. 1 for function classes beyond that in Assumption 2.

**Theorem 3** (Regret for linear functions with Option $I$). *If the function class is linear, i.e., $f_\theta(x, a) = \theta^\top \phi(x, a)$, and $\Theta = \{\|\theta\| < \sqrt{d}\}$, $\|\phi(x, a)\| < \sqrt{d}$. Then, applying Alg. 1 with Option I to discretized parameter space $\Theta^\epsilon$ which is an $\epsilon$-mesh. Let $\epsilon = 1/T$, $\alpha = 1$ and $\eta_{t,a} = \frac{1}{2}\sqrt{\beta_t \phi_{t,a} \Lambda^{-1} \phi_{t,a}}$ where $\Lambda_t = I + \sum_{t' < t} \phi_{t'} \phi_{t'}$, $\phi_{t,a} = \phi(x_t, a)$, $\phi_t = \phi_{t,a_t}$, $\beta_t = \max(128 d \ln t \ln(t^2/\delta), (\frac{8}{3} \ln(t^2 \delta))^2)$ is the confidence width in (Dani et al., 2008), we have with probability at least $1 - 2\delta$, the regret of Alg. 1 with Option I satisfies*

$$R_T \le \sqrt{8d^2 T \beta_T \ln T} + \sqrt{d} + 32\sqrt{d^2 T \beta_T} \log \frac{T}{\delta} = \tilde{O}(d\sqrt{T}).$$

The $\epsilon$-mesh in Theorem 3 introduces an addition regret than standard LinUCB. More importantly, the calculation of $\eta_{t,a}$ is expensive. We can improve the performance on linear functions with Option II which invokes the optimization oracle two times for each action per round.

We postpone the proof of Theorem 3 to Appendix.

**Theorem 4** (Regret for linear functions with Option II). *Let $\bar{\theta}_t = \arg\min_\theta MSE(\theta; D_t) + \|\theta\|^2/|D_t|$ and $\hat{\theta}_{t,a} = \arg\max_\theta f_\theta(x_t, a) - \eta_{t,a}(MSE(\theta; D_t) + \|\theta\|^2)$. Let*

$$\text{Lin-U}(x_t, a) = f_{\bar{\theta}_{t-1}}(x_t, a) + \sqrt{f_{\hat{\theta}_{t,a}}(x_t, a) - f_{\bar{\theta}_{t-1}}(x_t, a)}. \tag{4}$$

*We have Lin-U$(x_t, a)$ is equivalent to upper confidence bound in LinUCB Abbasi-Yadkori et al. (2011). Thus, setting $1/(2\eta_{t,a})$ to be the confidence width in (Abbasi-Yadkori et al., 2011), we have with probability at least $1 - \delta$, the regret of Alg. 1 with Option II is*

$$R_T \le \tilde{O}(d\sqrt{T}).$$

*Proof.* LinUCB (Abbasi-Yadkori et al., 2011) uses the following upper confidence bound for $\phi(x_t, a)$.

$$\text{LinUCB}(x_t, a) = \bar{\theta}_{t-1}^\top \phi_{t,a} + \sqrt{\beta_t \phi_{t,a} \Lambda_t^{-1} \phi_{t,a}}. \tag{5}$$

where $\Lambda_t = I + \sum_{t' < t} \phi_{t'} \phi_{t'}^\top$, $\phi_{t,a} = \phi(x_t, a)$, $\phi_t = \phi(x_t, a_t)$ and $\beta_t$ is confidence width.

Then the proof is straightforward after observing that $\bar{\theta}_{t-1} = \Lambda_t^{-1} \sum_{t' < t} \phi_{t'} r_{t'}$ and $\hat{\theta}_{t,a} = \bar{\theta}_{t-1} + \Lambda_t^{-1} \phi_{t,a}/(2\eta_{t,a})$ and, thus, $(\hat{\theta}_{t,a} - \bar{\theta}_{t-1})^\top \phi_{t,a} = \frac{1}{2\eta_{t,a}} \phi_{t,a}^\top \Lambda_t^{-1} \phi_{t,a}$. □

### 3.2 AN EMPIRICALLY EFFICIENT GRADIENT DESCENT-BASED OPTIMIZATION ORACLE

In order to apply Alg. 1, we have to efficiently solve or approximate Eq. (2). In this section, we consider the case that Eq. (2) is differentiable. Thus, naturally, one can use gradient descent methods to approximately solve Eq. (2). However, it is still not manageable to optimize from scratch for every $(x_t, a)$. We propose to optimize from $\bar{\theta}_{t-1}$ for $(x_t, a)$.

More specifically, we approximately solve Eq. (2) by executing a few steps of gradient ascent starting from $\bar{\theta}_{t-1}$. That is, $\hat{\theta}_{j+1} = \hat{\theta}_j + \kappa \nabla_\theta (f_\theta(x_t, a) - \eta \mathcal{R}(\theta; D))|_{\theta=\hat{\theta}_j}$ with $\hat{\theta}_0 := \bar{\theta}_{t-1}$, where $\kappa$ is the step size. The above implementation is summarized in Alg. 2.

Alg. 2 essentially performs a local search around $\bar{\theta}_{t-1}$. Indeed Alg. 2 brings extra benefit for the optimization by starting from $\bar{\theta}_{t-1}$ in this case. This is because intuitively $\bar{\theta}_{t-1}$ often gets closer to $\hat{\theta}(\cdot, a)$ when $t$ is larger. For example, in linear contextual bandits, $\hat{\theta}_{t,a} - \bar{\theta}_{t-1} = \frac{1}{2\eta_{t,a}} \Lambda_t^{-1} \phi_{t,a}$, and $\|\frac{1}{2\eta_{t,a}} \Lambda_t^{-1} \phi_{t,a}\|$ monotonically decreases as $t$ becomes larger according to the definition of $\eta_{t,a}$.

It is easy to see that the time complexity of Alg. 2 is $O(Mp)$ where $p$ is the number of parameters. In Sec. 5, we can see that the regret of ROFU is low in practice with very small $M$.

---

**Algorithm 2** An efficient implementation to estimate $U(x_t, a)$

---
1: **Input**: Dataset $D_{t-1}$, $\bar{\theta}_{t-1} = \arg\min_\theta \mathrm{MSE}(\theta; D_{t-1}) + \|\theta\|^2/|D_t|$ and context-action pair $x_t, a$. Learning rate $\kappa$ and training steps $M$, hyperparameters $\eta$ and $\alpha$.
2: Set $\hat{\theta}_0(x_t, a) := \bar{\theta}_{t-1}$.
3: **for** $j = 1, ..., M$ **do**
4:     $\hat{\theta}_j(x_t, a) := \hat{\theta}_{j-1}(x_t, a) + \kappa\tilde{\nabla}_\theta(f_\theta(x_t, a) - \eta(\mathrm{MSE}(\theta; D) + \alpha\|\theta\|^2))|_{\theta=\hat{\theta}_{j-1}(x_t, a)}$, where $\tilde{\nabla}$ is an estimator of gradient.
5: **end for**
6: **return** $\hat{\theta}_M(x_t, a)$.

---

### 3.3 PROOF OF THEOREM 1

Let us start with some useful notations. For convenience, we abbreviate $\eta_{t,a}$ as $\eta_t$ since $\eta_{t,a} = \eta_{t,a'}, \forall a, a'$ in Theorem 1. Let $\hat{\theta}_{t,a} = \arg\max_{\theta\in\Theta} f_\theta(x_t, a) - \eta_t\mathrm{MSE}(\theta; D)$ denote the parameter used in $U(x_t, a)$, $\Delta_t(\theta) = (f_\theta(x_t, a_t) - r_t)^2 - (f_{\theta^*}(x_t, a_t) - r_t)^2$ and $\lambda_t(\theta) = (f_\theta(x_t, a_t) - f_{\theta^*}(x_t, a_t))^2$. It is easy to see that $\mathbb{E}\Delta_t(\theta) = \lambda_t(\theta)$ where the expectation is over the randomness of the reward.

Lemma 2 presents a consentration inequality of $\sum\Delta_t(\theta)$ which is derived from Lemma 1.

**Lemma 1** (Freedman-type inequality, Theorem 1 in (Beygelzimer et al., 2011)). *For any martingale $Z_t$ with $|Z_t| \leq R$, with probability at least $1 - \delta$, for $\alpha \in (0, 1/R]$,*

$$\sum_{t=1}^{T} Z_t \leq \sum_{t\leq T} \alpha(e-2)\mathbb{E}[Z_t^2] + \frac{R\log 1/\delta}{\alpha}. \tag{6}$$

**Lemma 2** (Bounded differences). *With probability at least $1 - \delta$, $\forall\theta \in \Theta, t \leq T$,*

$$2\sum_{t'\leq t}\Delta_t(\theta) + 16\log\frac{|\Theta|T}{\delta} > \sum_{t'\leq t}\lambda_t. \tag{7}$$

*Proof.* This lemma is essentially a restatement of Lemma 4 in (Foster et al., 2018). Applying Lem 1 to martingale $\{\mathbb{E}\Delta_t(\theta) - \Delta_t(\theta)\}_{t\leq T}$, we have with probability at least $1 - \delta/(|\Theta|T)$

$$\sum_{t\leq T}\mathbb{E}\Delta_t(\theta) - \Delta_t(\theta) \leq \alpha(e-2)\sum_{t\leq T}\mathbb{E}[(\mathbb{E}\Delta_t(\theta) - \Delta_t(\theta))^2] + \frac{\log(|\Theta|T/\delta)}{\alpha}$$

$$\leq 4\alpha(e-2)\sum_{t\leq T}\mathbb{E}\Delta_t(\theta) + \frac{\log(|\Theta|T/\delta)}{\alpha}$$

The second inequality is because $\mathbb{E}[(\mathbb{E}\Delta_t(\theta) - \Delta_t(\theta))^2] \leq 4\mathbb{E}\Delta_t(\theta)$. Setting $\alpha = 1/8$ and rearranging, with fact $\lambda_t(\theta) > 0$ and applying union bound, we have with probability at least $1 - \delta$

$$\forall t \leq T, \theta \in \Theta, 2\sum_{t'\leq t}\Delta_t(\theta) + 16\log\frac{|\Theta|T}{\delta} > \sum_{t'\leq t}\lambda_t$$

$\square$

Let $a_t^* = \arg\max_a f_{\theta^*}(x_t, a)$, according to the definition of $\hat{\theta}_{t,a}$ and the definition of $a_t$, we have

$$f_{\theta^*}(x_t, a_t^*) - \eta_t\mathrm{MSE}(\theta^*) \leq f_{\hat{\theta}_{t,a_t^*}}(x_t, a_t^*) - \eta_t\mathrm{MSE}(\hat{\theta}_{t,a_t^*}),$$

$$f_{\hat{\theta}_{t,a_t^*}}(x_t, a_t^*) - \eta_t\mathrm{MSE}(\hat{\theta}_{t,a_t^*}) \leq f_{\hat{\theta}_{t,a_t}}(x_t, a_t) - \eta_t\mathrm{MSE}(\hat{\theta}_{t,a_t}).$$

Summing up the above two inequalities leads to

$$f_{\theta^*}(x_t, a_t^*) \le f_{\hat{\theta}_{t, a_t}}(x_t, a_t) - \eta_t(\text{MSE}(\hat{\theta}_{t, a_t}; D_t) - \text{MSE}(\theta^*; D_t)). \tag{8}$$

Now we present the proof of Theorem 1.

*Proof of Theorem 1.* Assuming the events in Lemma 2 happen, we have

$$\begin{aligned}
R_T &= \sum_{t \le T} f_{\theta^*}(x_t, a_t^*) - f_{\theta^*}(x_t, a_t) \\
&\le \sum_{t \le T} f_{\hat{\theta}_{t, a_t}}(x_t, a_t) - \eta_t(\text{MSE}(\hat{\theta}_{t, a_t}; D_t) - \text{MSE}(\theta^*; D_t)) - f_{\theta^*}(x_t, a_t) \\
&\le \sum_{t \le T} f_{\hat{\theta}_{t, a_t}}(x_t, a_t) - f_{\theta^*}(x_t, a_t) - \eta_t/(t-1) \sum_{t' < t} \lambda_{t'}(\hat{\theta}_{t, a_t}) + \eta_t 16/(t-1) \log \frac{|\Theta|T}{\delta} \\
&\le \sum_{t \le T} \left( \frac{16\eta_t}{t-1} \log \frac{|\Theta|T}{\delta} + c_2 g(\theta) - \eta_t c_1^2 g(\theta)^2 \right) \\
&\le \sum_{t \le T} \left( \frac{16\eta_t}{t-1} \log \frac{|\Theta|T}{\delta} + \frac{c_2^2}{4\eta_t c_1^2} \right) \\
&\le \sum_{t \le T} 4\sqrt{\frac{c_2^2}{(t-1)c_1^2} \log \frac{|\Theta|T}{\delta}} \\
&\le 8 \frac{c_2}{c_1} \sqrt{T \log \frac{|\Theta|T}{\delta}}
\end{aligned}$$

The first line is the definition of regret; the second line is according to Eq. (8); the third line is by Lemma 2; the fourth line is by Assumption 2; the second last line is by setting $\eta_t = \frac{1}{8} \sqrt{\frac{c_2^2(t-1)}{c_1^2 \log \frac{|\Theta|T}{\delta}}}$.

$\square$

## 4 RELATED WORK

**Optimism in face of uncertainty** (OFU): Our algorithm is essentially a variant of OFU principle, which is a powerful framework to trade-off exploration and exploitation for bandit problems. As discussed in Section 3, OFU algorithms take actions according to an optimistic estimation over the reward. Most OFU algorithms optimistically estimates the reward by the best possible value over a confidence set of the reward functions, i.e.,

$$\text{OFU}^S(x, a) := \max_{\theta \in \Theta_\delta} f_\theta(x, a), \tag{9}$$

where $\Theta_\delta := \{\theta : \mathbb{P}(D|\theta) > \delta\}$ and $\mathbb{P}(D|\theta)$ is the likelihood of $D$ given $\theta$. In many cases, the constraint can be replaced by $\text{MSE}(\theta; D) < \delta$. For simple function classes such as multi-armed bandit and linear contextual bandit, $\text{OFU}^S$ has closed-form solutions [1].

However, for more complex tasks, OFU algorithms explicitly maintain a confidence set, e.g., see (Foster et al., 2018; Dudik et al., 2011). The cost of explicitly maintaining a confidence set is extremely expensive for complex problems or function classes, such as deep neural networks. One alternative approach is to consider the Lagrangian Multiplier method, i.e., solving $\min_\theta \max_\eta f_\theta(x, a) - \eta \text{MSE}(\theta; D)$. But this is still much slower than our optimization oracle in general.

Our results, theorem 1 and theorem 4, suggest that we can effectively trade-off exploration and exploitation while avoiding explicitly maintaining the confidence set.

---

[1]The computational cost of closed-form solutions can be expensive for high-dimensional problems even when the function class is linear.

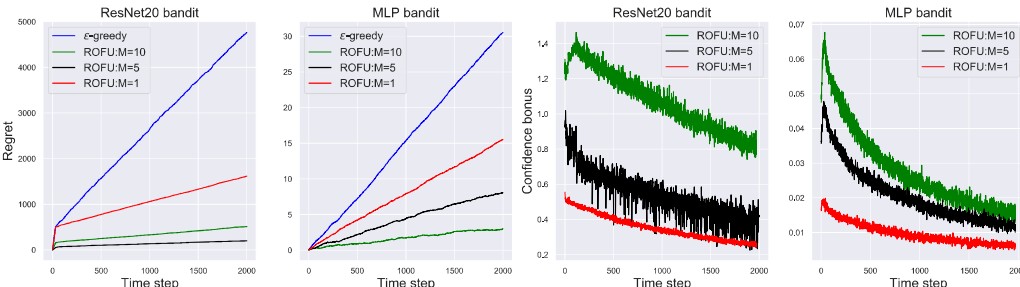

Figure 1: Ablation study on MLP and ResNet bandits. Notation: $M = 1/5/10$ means that we run $1/5/10$ gradient descent updates in Alg. 2.

**Contextual bandit algorithms with optimization oracle**: As mentioned in section 1, some contextual bandit algorithms invoke optimization oracles to accelerate computation. Agarwal et al. (2014); Dudik et al. (2011) rely on cost-sensitive policy classification oracles and achieve an optimal regret of $O(\sqrt{T \log(T|\Pi|/\delta)})$ where $\Pi$ is the space of policies. This kind of oracles can be computational inefficient for complex function classes. And these algorithm call the optimization algorithm many times in each round to achieve a regret guarantee, e.g. , Dudik et al. (2011) calls the optimization oracle for $O(T^5)$ times. Foster et al. (2018) access a regression oracle which is special case of Eq. (2) with $\eta \approx +\infty$. But Foster et al. (2018) calls the oracle for $O(\log T)$ times for each action in each round. More importantly, they explicitly maintain a subset of $\Theta$ with $\mathrm{MSE}(\theta; D) < \min_{\theta'} \mathrm{MSE}(\theta'; D) + \beta$. And as discussed above, maintaining such confidence set is infeasible for complex function classes.

**Algorithms for deep contextual bandit**: We note that there are attempts (Zhou et al., 2020; Zhang et al., 2020) to extend the realizability-based algorithms to deep neural networks. NeuralUCB (Zhou et al., 2020) and Neural Thompson sampling (Zhang et al., 2020) conduct experiments on multilayer neural network with significantly simplified and approximate implementation to accelerate computation. The analyses do not apply to general function classes.

## 5 EXPERIMENT

We now empirically evaluate ROFU. We only present empirical results for Option II as the empirical performances of Option II is slightly better than Option $I$. For simplicity, in all our experiments, we set $\eta_{t,a} = 1$ and $\alpha = 1$. More specifically, we train $\bar{\theta}_{t-1}$ by minimizing $\mathrm{MSE}(\theta; D_t)$ with standard optimizer [2] and $\hat{\theta}_{t,a}$ is trained to maximize $f_\theta(x, a) - |D|\mathrm{MSE}(\theta; D)$ using Alg. 2. And $\mathrm{OFU}^R(x_t, a) = f_{\bar{\theta}_{t-1}}(x_t, a) + \sqrt{f_{\hat{\theta}_{t,a}}(x_t, a) - f_{\bar{\theta}_{t-1}}(x_t, a)}$ as in Option II.

### 5.1 ANALYSIS ON MLP AND RESNET BANDITS

The goal of this work is to develop a contextual bandit algorithm which is efficient in trading off EE when reward is generated from a complex function while keeping a low cost on computational resources. From Alg. 2, we can see that the time complexity of ROFU is determined by the training step $M$. To reduce computational cost, we evaluate ROFU when $M$ is small in experiments. As suggested by the experimental results, setting $M$ to be a relatively small value doesn't hurt the performance much.

To evaluate the performance of ROFU in Alg. 2 on complex tasks, we consider two contextual bandits with a DNN as the simulator. That is, $r(x_t, a)$ is generated by a DNN model. We consider two popular DNN architectures to generate rewards: 2-layer MLP and 20-layer ResNet with CNN blocks and Batch Normalization as in He et al. (2016). We summarize other information of the two tested bandits in Table. 1.

---

[2]We train $\bar{\theta}_{t-1}$ with stochastic gradient descent starting from $\bar{\theta}_{t-2}$ in all the experiments.

| Bandit | Layer | Context Dim | # Arms | NN Parameters | Context Distribution | Noise |
|--------|-------|-------------|--------|---------------|---------------------|-------|
| MLP | 2 | 10 | 10 | Random | Gaussian | $\mathcal{N}(0, 0.05)$ |
| ResNet | 20 | $3 \times 32 \times 32$ | 10 | Trained on Cifar10 | Uniform | $\mathcal{N}(0, 0.5)$ |

Table 1: Basic information about MLP and ResNet bandits.

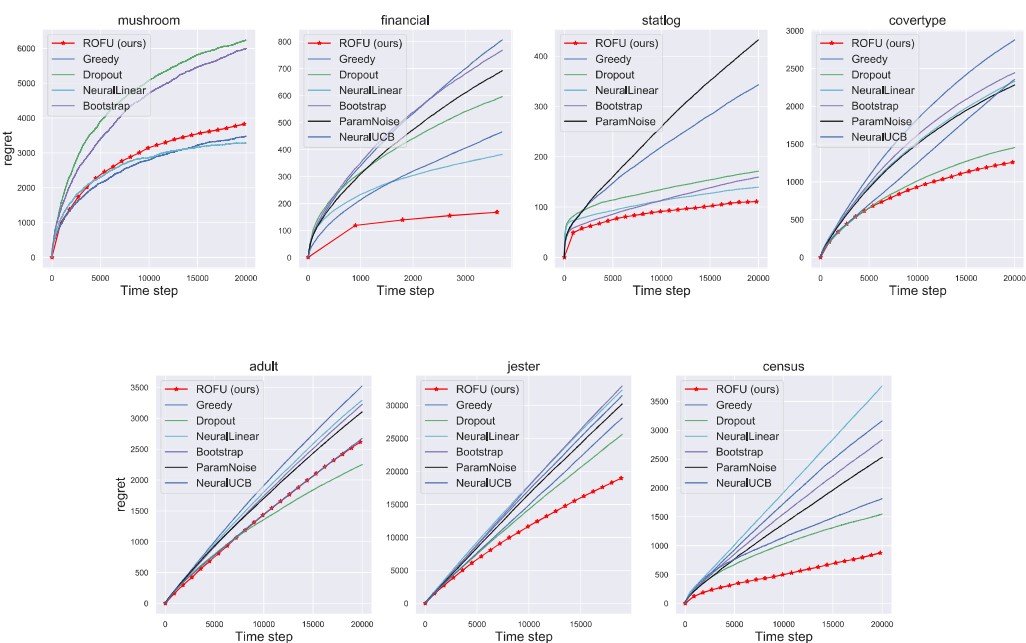

Figure 2: Evaluations on non-linear contextual bandits.

We use DNNs with larger size for training in Alg. 2. More specifically, for MLP-bandit, $f_\theta$ is chosen as a 3-layer MLP and for ResNet20-bandit, $f_\theta$ is chosen as ResNet32. Each experiment is repeated for 16 times. We present the regret and confidence bonus in Fig. 1. From Fig. 1, we can see that (1) ROFU can achieve a small regret on both tasks with a considerably small $M$ even for very large DNN model; (2) The confidence bonus monotonically increases with $M$. For each $M$, the confidence bonus converges to 0 as expected. Moreover, while the regret seems sensitive to the value of $M$, the regrets of ROFU with $M = 5, 10$ are much smaller than the case of $M = 1$ and $\epsilon$-greedy.

## 5.2 PERFORMANCE COMPARISON ON REAL-WORLD DATASETS

To evaluate ROFU against powerful baselines, we conduct experiments on contextual bandits which are created from real-world datasets, following the setting in Riquelme et al. (2018). For example, suppose that $D := \{(x_t, c_t)\}_{t \leq T}$ is a $K$-classification dataset where $x_t$ is the feature vector and $c_t \in [K]$ is the label. We create a contextual bandit problem as follows: at time step $t \leq T$, the agent observes context $x_t$, and then takes an action $a_t$. The agent receives high reward if it successfully predicts the label of $x_t$. For non-classification dataset, we can turn it into contextual bandit in similar ways. For the details of these bandits, please refer to Riquelme et al. (2018).

For baselines, we consider NeuralUCB Zhou et al. (2020) and Thompson sampling variants from Riquelme et al. (2018). It is noteworthy that we only evaluate the algorithms in Riquelme et al. (2018) with relatively small regrets. We directly run the code provided by the authors. For ROFU, we fix $M = 5$ for all experiments. We tune other hyper-parameters of ROFU on statlog and directly apply the hyperparameters on statlog to other datasets except mushroom. This is because the reward scale of mushroom is much larger than other datasets. For baselines, we directly use the best reported hyper-parameters.

We report the regret $R_T = \mathbb{E} \sum_{t \leq T} r(x_t, a_t') - \mathbb{E} \sum_{t \leq T} r(x_t, a_t)$ where $a_t' = \arg\max_{a \in A} f_{\theta'}(x_t, a)$ and $\theta'$ is the parameter trained by minimizing MSE on the whole dataset. The results are presented in Fig. 2 and Table 2. We found that the regret of NeuralUCB is occasionally linear. This might be because that NeuralUCB uses a diagonal matrix to approximate $Z$ to accelerate. Moreover, we can see that ROFU significantly outperforms these baselines in terms of regret.

| | Mean | Census | Jester | Adult | Covertype | Statlog | Financial | Mushroom |
|---|---|---|---|---|---|---|---|---|
| Dropout | $1.75_{\pm 0.80}$ | $1.51_{\pm 0.10}$ | $1.34_{\pm 0.14}$ | $\mathbf{1.00}_{\pm \mathbf{0.09}}$ | $1.14_{\pm 0.13}$ | $1.54_{\pm 0.87}$ | $3.50_{\pm 0.60}$ | $2.21_{\pm 0.42}$ |
| Bootstrap | $2.23_{\pm 1.00}$ | $2.51_{\pm 0.16}$ | $1.72_{\pm 0.11}$ | $1.43_{\pm 0.10}$ | $1.93_{\pm 0.13}$ | $1.43_{\pm 1.57}$ | $4.52_{\pm 2.29}$ | $2.04_{\pm 0.48}$ |
| ParamNoise | $2.30_{\pm 1.12}$ | $2.28_{\pm 0.23}$ | $1.59_{\pm 0.14}$ | $1.37_{\pm 0.10}$ | $1.80_{\pm 0.20}$ | $3.88_{\pm 6.40}$ | $4.07_{\pm 1.76}$ | $1.06_{\pm 0.32}$ |
| NeuralLinear | $1.82_{\pm 0.69}$ | $3.24_{\pm 0.47}$ | $1.70_{\pm 0.13}$ | $1.46_{\pm 0.12}$ | $1.84_{\pm 0.19}$ | $1.25_{\pm 0.11}$ | $2.25_{\pm 0.35}$ | $\mathbf{1.00}_{\pm \mathbf{0.38}}$ |
| Greedy | $2.47_{\pm 1.12}$ | $2.76_{\pm 1.24}$ | $1.65_{\pm 0.10}$ | $1.56_{\pm 0.11}$ | $2.27_{\pm 0.23}$ | $3.08_{\pm 4.91}$ | $4.74_{\pm 2.31}$ | $1.20_{\pm 0.41}$ |
| NeuralUCB | $9.76_{\pm 14.02}$ | $1.72_{\pm 0.12}$ | $1.47_{\pm 0.08}$ | $1.18_{\pm 0.05}$ | $1.86_{\pm 0.16}$ | $41.42_{\pm 69.51}$ | $2.74_{\pm 0.50}$ | $17.29_{\pm 7.45}$ |
| ROFU (ours) | $\mathbf{1.05}_{\pm \mathbf{0.09}}$ | $\mathbf{1.00}_{\pm \mathbf{0.09}}$ | $\mathbf{1.00}_{\pm \mathbf{0.20}}$ | $1.17_{\pm 0.06}$ | $\mathbf{1.00}_{\pm \mathbf{0.14}}$ | $\mathbf{1.00}_{\pm \mathbf{0.24}}$ | $\mathbf{1.00}_{\pm \mathbf{0.60}}$ | $1.22_{\pm 0.37}$ |

Table 2: The final regret of each algorithm. The regrets are normalized according to the algorithm with smallest regret.

## CONCLUSION AND FUTURE WORK

In this work, we propose an OFU variant, called ROFU, which is designed for contextual bandit with general function classes. We show that the regret of ROFU is (near-)optimal under very mild assumptions. Moreover, we propose an efficient algorithm to approximately compute the upper confidence bound. Thus, ROFU is efficient in both computation and EE trade-off, which are empirically verified by our experimental results.

EE trade-off is a fundamental problem that lies in the heart of sequential decision making. However, the huge computational cost of (near-)optimal EE trade-off algorithms significantly limits the application, especially on complex domain. We hope our method could inspire more algorithms to efficiently trade-off EE for sequential decision-making tasks beyond contextual bandit, such as deep reinforcement learning.

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
