# OpenReview forum: "Regularized-OFU: an efficient algorithm for general contextual bandit with optimization oracles"
_ICLR.cc/2022/Conference — ICLR 2022 Submitted_

### Official Review · Reviewer_sNy5 · 2021-10-30

**Correctness:** 3
**Technical Novelty And Significance:** 2
**Empirical Novelty And Significance:** 3
**Recommendation:** 5
**Confidence:** 4

**Main Review:**

Contextual bandit with general function classes is an interesting problem in machine learning. However, I think this paper still has some concerns.

1. Thm 1 requires the parameter space $\Theta$ to be a finite set. However, it is not clear in the paper how to deal with the finite parameter space. For example, the authors say max can be solved using a gradient-based algorithm (right after Definition 2), but don't show how to adjust the gradient-based algorithm to finite parameter space.

2. I think the conclusion of Thm 3 should be $R_T\leq \tilde{O}(d\sqrt{dT})$ because $\beta_T=\tilde{O}(d)$.

3. When the authors claim Assumption 2 is mild (right after Assumption 2), they say if the h function is smooth. However, the absolute value function usually affects smoothness (by at least one point).

4. What's the intuition of Option II? I suggest the authors provide more discussion on Option II and introduce the relationship between Option I and Option II.

5. I'm confused with the proof of Thm 2. Thm 2 proves the lower bound by giving an instance of the linear function. However, as presented in the second line of page 4, the linear function doesn't satisfy assumption 2. Thus the given instance doesn't meet the condition of Thm 2.

6. In section 5, the authors only present the results of Option II. However, I think it would be better to show the results of both Option I and Option II because Option I has regret bound for general functions. Also, I'm curious about whether Option II also has regret bound for general functions since the paper claims that Option II performs better than Option I.

7. Some experimental results haven't converged.

**Summary Of The Paper:**

This paper studies contextual bandits with general function classes. The authors propose Regularized OFU algorithm where the agent selects the action maximizing the OFU function in each round. The proposed methods achieves $O(\sqrt{T\log\frac{|\Theta|T}{\delta}})$ regret for general functions under mild conditions and achieves $\tilde{O}(d\sqrt{T})$ regret for linear functions.

**Summary Of The Review:**

I think the theoretical part is not very consistent with the motivation and the experiments. It is not very clear why to consider such two options and why the experiments only use option II but in non-linear case.

---

### Official Review · Reviewer_aMkn · 2021-11-02

**Correctness:** 2
**Technical Novelty And Significance:** 2
**Empirical Novelty And Significance:** 3
**Recommendation:** 3
**Confidence:** 4

**Main Review:**

I think the biggest issue of this paper to me is Assumption 2. The authors keep saying that this assumption is "very mild", but they don't even give a concrete example that satisfy this condition. To me this is a very strong assumption. It means that if $\theta$ is not optimal, $f_\theta$ is strictly worse than the optimal value at every (x,a). Even the simplest case (linear function) I can think of does not satisfy this assumption. I think this assumption makes sense only if c1 is very very small, but it makes the regret bounds vacuous.

The theorem is also strange to me. Theorem 1 only shows the case when $\alpha=0$. If so, why do we even need the regularization term in the algorithm? I didn't find any discussion in the experiments, either.

As for the algorithm design, I think it is reasonable and I appreciate its simplicity. However, I don't think authors do a sufficient literature review. The citations are relatively few. Although I cannot come up with any previous work that consider similar algorithms, I think the authors should compare more previous methods to support the novelty of their algorithm. For example, with some random search on the Internet, I found a paper "Regularized Contextual Bandits" by Fontaine et al. I am not saying the paper is comparable to this work but I think this is something the authors should clarify. Moreover, the survey should not only cover contextual bandits but discuss related fields such as reinforcement learning and online regression literatures.


**Summary Of The Paper:**

The authors present an algorithm called regularized optimism in face of uncertainty (ROFU) for general contextual bandit problems. It extends upper confidence bounds to general functions. When the optimization oracle can be efficiently solved, the algorithm can be efficiently implemented. Some theoretical regret bounds and experiments are provided.

**Summary Of The Review:**

The strong assumption makes theoretical contributions of this paper incremental. The authors also didn't do enough literature survey to support the novelty of their algorithm. Therefore, I tend to recommend rejection.

---

### Official Review · Reviewer_xPt8 · 2021-11-02

**Correctness:** 3
**Technical Novelty And Significance:** 1
**Empirical Novelty And Significance:** 2
**Recommendation:** 3
**Confidence:** 4

**Main Review:**

Strengths:
The paper conducts numerical experiments and shows that ROFU outperforms many popular baselines including the UCB and Thompson sampling variants when using deep neural networks as the function class. The experimental results are interesting.

Weaknesses:
- The paper claims that "theoretically, for general function classes under very mild assumptions, ROFU achieves a near-optimal regret bound" in the abstract, and highlights it as a main contribution throughout the paper. I find this claim misleading. The regret bound for general function classes is stated in Theorem 1, which relies on a super strong assumption — Assumption 2, for which the authors do not provide any concrete and practical function classes as examples. Importantly, Assumption 2 does not even hold for linear function classes. This makes the use of the term "general contextual bandits" (in the title and main body) very misleading — in fact, Assumption 2 is far more restrictive than the assumption of linear function classes, and I believe that this assumption can directly make exploration unnecessary (i.e., *the greedy algorithm can directly achieves the near-optimal $\tilde{O}(\sqrt{T})$ regret bound*). Since there is no exploration-exploitation trade-off under Assumption 2 at all, I don't feel that Theorem 1 can be viewed as a non-trivial result for "general contextual bandits".
  - Let me explain why Assumption 2 is strong in more detail. Assumption 2 essentially says that, for arbitrary $x$ and $a$, $|f_{\theta}(x,a)-f_{\theta^*}(x,a)|=\Theta(\textrm{distance}(\theta,\theta^*))$. This means that the choice of actions $a_1,a_2,\dots,a_t$ is not important for learning $\theta^*$ at all, as any sequence of $(x_1,a_1),(x_2,a_2),\dots,(x_t,a_t)$ always gives the same estimation rate of $\textrm{distance}(\theta,\theta^*)$ by running least squares regression. Moreover, $\textrm{distance}(\theta,\theta^*)$ can also directly control the regret bound. Therefore, it seems that even the greedy algorithm can achieve near-optimal regret under Assumption 2.
  - Although the paper also has other theoretical results (Theorem 3 and Theorem 4) on the regret bounds for linear function classes, the fact that OFU-type algorithms can attain near-optimal regret for linear function classes is already well-known. Therefore, it is not clear what is the contribution of Theorem 3 and Theorem 4 .

- Even if the authors can show Theorem 1 under much weaker assumptions on the function class, it is unclear to me whether the algorithm really has computational advantages over some existing algorithms based on the least squares oracle. Specifically, the optimization oracle assumed in the current paper (see Definition 2) seems to be strictly more restrictive than the least squares oracles used in [1,2,3]. It is also worth noting that [2] and [3] have already provided optimal and efficient algorithms for general stochastic contextual bandits using the least squares oracle (under no assumption on the function class and standard/weak assumptions on the action space). Therefore, even if the authors can extend Theorem 1 to a much more general setting, comparisons to existing literature [2,3] seem to be important.
  - By the way, in Section 4, when the authors compare their result to [1], the authors say that "[1] explicitly maintain a subset of $\Theta$ ...". To my knowledge, this statement is not correct — [1] does not explicitly maintain the subset, but designs a oracle-efficient algorithm to cleverly compute the confidence bounds (see Theorem 1 therein).

- There are several unclear places in the paper. For example, the "MSE" appears in Definition 2 is not really defined in the main article. Also, in Algorithm 2, it is unclear to me how $\tilde{\Delta}$, the estimator of gradient, can be efficiently computed using the assumed optimization oracle.

References:

[1] Foster, D., Agarwal, A., Dudík, M., Luo, H. and Schapire, R., 2018, July. Practical contextual bandits with regression oracles. In International Conference on Machine Learning (pp. 1539-1548). PMLR.

[2] Simchi-Levi, D. and Xu, Y., 2020. Bypassing the monster: A faster and simpler optimal algorithm for contextual bandits under realizability. Available at SSRN 3562765.

[3] Xu, Y. and Zeevi, A., 2020. Upper counterfactual confidence bounds: a new optimism principle for contextual bandits. arXiv preprint arXiv:2007.07876.


**Summary Of The Paper:**

The paper presents a contextual bandit algorithm called regularized optimism in the face of uncertainty (ROFU) which is designed for general function classes. The algorithm is claimed to achieve near-optimal regret "for general functions under very mild assumptions". However, I find this claim misleading. The optimization oracle considered in the paper also seems to be more restrictive than the least squares oracle considered in recent literature.

The paper also conducts numerical experiments to examine the empirical performance of the ROFU algorithm and compare it to several baselines. The performance of the ROFU algorithm seems promising. I feel that the experimental part of the paper is more solid and interesting than the theoretical part.

**Summary Of The Review:**

The experimental results of the paper may have potential. However, the existing theoretical results are relatively weak and overclaimed. The paper also fails to demonstrate its contributions compared with prior works. Therefore, I do not think that the paper can be accepted in its current form.

---

### Official Review · Reviewer_e8Nb · 2021-11-07

**Correctness:** 3
**Technical Novelty And Significance:** 3
**Empirical Novelty And Significance:** 2
**Recommendation:** 5
**Confidence:** 4

**Main Review:**

Strengths:
I think the paper is tackling an important aspect of the contextual bandit problem, aiming for flexible function classes and efficient implementation. The idea of utilizing regression oracles has been proposed previously, but the authors focused on designing an algorithm that is efficiently implementable for practical usage.

The generalization of linear bandit is also nice and makes sense.

Weaknesses:
First, the authors argue that "the availability of such optimization oracle [for solving Eq.(2)] is a very mild assumption in practice." In practice, there are algorithms to "approximately" solve Eq.(2). To exactly solve for possible non-convex function $f_\theta$ is not mild. The regret analysis is based on exactly solve Eq.(2). If one allows for approximate solutions for the optimization problem that the paper presents, then it would be fair to allow for similar approximate solutions for the previous works that the authors criticize for not having tractable solutions.

Another drawback is the lack of comparisons with more recent work including  Foster & Rakhlin, (2020) and Simchi-Levi & Xu (2020). The authors mention Foster & Rakhlin, (2020) once in the introduction but do not directly compare with their algorithm and results despite the fact that it is also a regression oracle-based algorithm with flexible function classes. The paper does not mention Simchi-Levi & Xu (2020) at all, which uses a regression oracle under realizability assumption.

#References
Simchi-Levi, David, and Yunzong Xu. "Bypassing the monster: A faster and simpler optimal algorithm for contextual bandits under realizability." Available at SSRN 3562765 (2020).


**Summary Of The Paper:**

The paper proposes a UCB variant called Regularized-OFU (ROFU) with provably near-optimal regret. ROFU computes UCBs with an optimization oracle which, the authors claim, can be efficiently implemented for flexible function classes.

The paper provides a sublinear regret bound of $O(\sqrt{T \log \frac{|\Theta|T}{\delta}})$ that matches the lower bound up to a logarithmic factor. The paper also provides experimental results to support their claims of efficient implementation.


**Summary Of The Review:**

With the reasons above, I am leaning towards rejection.

---

### Decision · Program_Chairs · 2022-01-20

**Decision:**

Reject

**Comment:**

This paper tackles the contextual bandit problem with general function classes and introduces a novel algorithm called regularized optimism in face of uncertainty (ROFU). Although this is an important and relevant problem, the theoretical contribution is rather weak due to the strong assumptions, which also results in a lack of consistency with the motivation and the empirical settings. Moreover, although experimental results suggest that the proposed ROFU method may have potential, the empirical contribution is unclear as the paper currently lacks a comparison with appropriate previous work. All these concerns were raised in the reviews, but unfortunately, none were addressed in the rebuttal phase.